# Dynamic Update-to-Data Ratio: Minimizing World Model Overfitting

**Nicolai Dorka[1]**  **Tim Welschehold[1]**  **Wolfram Burgard[2]**
[1]University of Freiburg  [2]University of Technology Nuremberg
dorka@cs.uni-freiburg.de

## Abstract

Early stopping based on the validation set performance is a popular approach to find the right balance between under- and overfitting in the context of supervised learning. However, in reinforcement learning, even for supervised sub-problems such as world model learning, early stopping is not applicable as the dataset is continually evolving. As a solution, we propose a new general method that dynamically adjusts the update to data (UTD) ratio during training based on under- and overfitting detection on a small subset of the continuously collected experience not used for training. We apply our method to DreamerV2, a state-of-the-art model-based reinforcement learning algorithm, and evaluate it on the DeepMind Control Suite and the Atari 100k benchmark. The results demonstrate that one can better balance under- and overestimation by adjusting the UTD ratio with our approach compared to the default setting in DreamerV2 and that it is competitive with an extensive hyperparameter search which is not feasible for many applications. Our method eliminates the need to set the UTD hyperparameter by hand and even leads to a higher robustness with regard to other learning-related hyperparameters further reducing the amount of necessary tuning.

## 1 Introduction

In model-based reinforcement learning (RL) the agent learns a predictive world model to derive the policy for the given task through interaction with its environment. Previous work has shown that model-based approaches can achieve equal or even better results than their model-free counterparts Silver et al. (2018); Schrittwieser et al. (2020); Chua et al. (2018); Hafner et al. (2021). An additional advantage of using a world model is, that once it has been learned for one task, it can directly or after some finetuning be used for different tasks in the same environment potentially making the training of multiple skills for the agent considerably cheaper. Learning a world model is in principle a supervised learning problem. However, in contrast to the standard supervised learning setting, in model-based RL the dataset is not fixed and given at the beginning of training but is gathered over time through the interaction with the environment which raises additional challenges.

A typical problem in supervised learning is overfitting on a limited amount of data. This is well studied and besides several kinds of regularizations a common solution is to use a validation set that is not used for training but for continual evaluation of the trained model during training. By considering the learning curve on the validation set it is easy to detect if the model is under- or overfitting the training data. For neural networks a typical behavior is that too few updates lead to underfitting while too many updates lead to overfitting. In this context, the validation loss is a great tool to balance those two and to achieve a small error on unseen data.

For learning a world model on a dynamic dataset there unfortunately is no established method to determine if the model is under- or overfitting the training data available at the given point in time. Additionally, in model-based RL a poorly fit model can have a dramatic effect onto the learning result as from it the agent derives the policy, which influences the future collected experience which again influences the learning of the world model. So far, in model-based RL this is commonly addressed with some form of regularization and by setting an update-to-data (UTD) ratio that specifies how many update steps the model does per newly collected experience, similar to selecting the total number of parameter updates in supervised learning. Analogously to supervised learning, a higher

UTD ratio is more prone to overfit the data and a lower one to underfit it. State-of-the-art methods set the UTD ratio at the beginning of the training and do not base the selection on a dynamic performance metric. Unfortunately, tuning this parameter is very costly as the complete training process has to be traversed several times. Furthermore, a fixed UTD ratio is often sub-optimal because different values for this parameter might be preferable at different stages of the training process.

In this paper, we propose a general method – called **D**ynamic **U**pdate-**t**o-**D**ata ratio (**DUTD**) – that adjusts the UTD ratio during training. DUTD is inspired by using early stopping to balance under- and overfitting. It stores a small portion of the collected experience in a separate validation buffer not used for training but instead used to track the development of the world models accuracy in order to detect under- and overfitting. Based on this, we then dynamically adjust the UTD ratio.

We evaluate DUTD applied to DreamerV2 Hafner et al. (2021) on the DeepMind Control Suite and the Atari100k benchmark. The results show that DUTD increases the overall performance relative to the default DreamerV2 configuration. Most importantly, DUTD makes searching for the best UTD rate obsolete and is competitive with the best value found through extensive hyperparameter tuning of DreamerV2. Further, our experiments show that with DUTD the world model becomes considerably more robust with respect to the choice of the learning rate.

In summary, this paper makes the following contributions: i) we introduce a method to detect under- and overfitting of the world model online by evaluating it on hold-out data; ii) We use this information to dynamically adjust the UTD ratio to optimize world model performance; iii) Our method makes tuning the UTD hyperparameter obsolete; iv) We exemplarily apply our method to a state-of-the-art model-based RL method and show that it leads to an improved overall performance and higher robustness compared to its default setting and reaches a competitive performance to an extensive hyperparameter search.

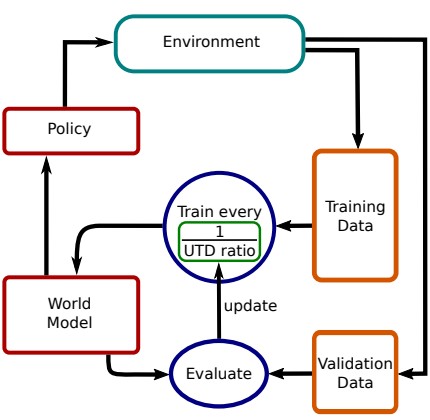

Figure 1: Overview of DUTD. A small subset of the experience collected from the environment is stored in a validation set not used for training. The world model is trained for one update after every $\frac{1}{UTD\ ratio}$ many environment steps. From time to time, *e.g.*, after an episode ended, the *UTD ratio* is adjusted depending on the detection of under- or overfitting of the world model on the validation data. The policy is obtained from the world model either by planning or learning and collects new data in the environment.

## 2 RELATED WORK

In reinforcement learning there are two forms of generalization and overfitting. Inter-task overfitting describes overfitting to a specific environment such that performance on slightly different environments drops significantly. This appears in the context of sim-to-real, where the simulation is different from the target environment on which a well performing policy is desired, or when the environment changes slightly, for example, because of a different visual appearance Zhang et al. (2018b); Packer et al. (2018); Zhang et al. (2018a); Raileanu et al. (2020); Song et al. (2020). In contrast, intra-task overfitting appears in the context of learning from limited data in a fixed environment when the model fits the data too perfectly and generalizes poorly to new data. We consider intra-task opposed to inter-task generalization.

In model-based reinforcement learning, there is also the problem of policy overfitting on an inaccurate dynamics model Arumugam et al. (2018); Jiang et al. (2015). As a result, the policy optimizes over the inaccuracies of the model and finds exploits that do not work on the actual environment. One approach is to use uncertainty estimates coming from an ensemble of dynamics models to be more conservative when the estimated uncertainty is high Chua et al. (2018). Another approach to prevent the policy from exploiting the model is to use different kinds of regularization on the plans the policy considers Arumugam et al. (2018). In contrast to these previous works, we directly

tackle the source of the problem by learning a better dynamics model. Consequently, our method is orthogonal to and can easily be combined with the just mentioned line of work.

Directly targeting the overfitting of the dynamics model can be done through the usage of a Bayesian dynamics model and the uncertainties that come with such a model. Gaussian processes have been used successfully in this context Deisenroth & Rasmussen (2011) although it is difficult to scale this to high-dimensional problems. Another way to reduce overfitting of the dynamics model is to use techniques from supervised learning. This includes for example regularization of the weights, dropout Srivastava et al. (2014), or data augmentation Laskin et al. (2020); Schwarzer et al. (2021). All of these are also orthogonal to our method and can be combined with it to learn an even better dynamics model. Another popular approach is early stopping Strand (1974); Anderssen & Prenter (1981); Morgan & Bourlard (1989), where the training is stopped before the training loss converges. Our method can be regarded as the analogy of early stopping in a dynamic dataset scenario.

Reducing the number of model parameters can prevent overfitting but can decrease performance compared to the right amount of training steps with more parameters. Our method overcomes this problem by automatically choosing the right amount of training steps for a given network.

Hyperparameter optimization for RL algorithms is also related to our work. For example, AlphaStar Silver et al. (2018) has been improved by using Bayesian optimization Chen et al. (2018). Zhang et al. (2021) demonstrated that model-based RL algorithms can be greatly improved through automatic hyperparameter optimization. A recent overview on automated RL is given by Parker-Holder et al. (2022). However, most of these approaches improve hyperparameters by training the RL agent on the environment in an inner loop while keeping the hyperparameters fixed during each run. Our work deviates from that by adapting a hyperparameter online during training of a single run. The approach of Schaul et al. (2019) also falls into this category and dynamically adapts behavior-related parameters such as stochasticity and optimism. Similarly, the algorithm Agent57 Badia et al. (2020) adaptively chooses from a set of policies with different exploration strategies and achievs human level performance on all $57$ Atari games Bellemare et al. (2013). Another approach adapts a hyperparameter that controls under- and overestimation of the value function online resulting in a model-free RL algorithm with strong performance on continuous control tasks Dorka et al. (2021).

In contrast to these approaches, our method directly learns a better world model by detecting under- and overfitting online on a validation set and dynamically adjusts the number of update steps accordingly. This renders the need to tune the UTD ratio hyperparameter unnecessary and further allows to automatically have its value being adapted to the needs of the different training stages.

## 3  THE DUTD ALGORITHM

In this section, we will first introduce the general setup, explain early stopping in the context of finding the right data fit and propose a new method that transfers this technique to the online learning setting. Lastly, we explain how the method can be applied to DreamerV2.

### 3.1  MODEL-BASED REINFORCEMENT LEARNING

We use the classical RL framework Sutton & Barto (2018) assuming a Markov decision process $(\mathcal{S}, \mathcal{A}, \mathcal{P}, \mathcal{R})$. In this framework, the agent sequentially observes the current state $s_t \in \mathcal{S}$ in which it executes an action $a_t \in \mathcal{A}$, receives a scalar reward $r_t$ according to the reward function $\mathcal{R}$, and transitions to a next state $s_{t+1}$ generated by the unknown transition dynamics $\mathcal{P}$. The goal is to learn a policy that selects actions in each state such that the total expected return $\sum_{i=t}^{T} r_i$ is maximized.

Model-based RL approaches learn a world model $\hat{\mathcal{P}}(s_{t+1} \mid s_t, a_t)$ – also called dynamics model – and a reward model $\hat{R}(r_t \mid s_t)$ that attempt to reflect their real but unknown counterparts. These models can then be used to learn a good policy by differentiating through the world model or by generating imaginary rollouts on which an RL algorithm can be trained. Alternatively, the learned model can be used in a planning algorithm to select an action in the environment.

### 3.2  UNDER- AND OVERFITTING

A well-known problem in supervised learning is that of overfitting, which typically corresponds to a low error on the training data and a high error on test data not seen during training. Usually, this

happens if the model fits the training data too perfectly. In contrast to this, underfitting corresponds to the situation in which the model even poorly fits the training data and is characterized by both a high training and test error. To measure the performance of the model on unseen data, the available data is often split into a training and a validation set. Generally, only the training set is used to train the model while the validation set is used to evaluate its performance on new data.

For iterative training methods – like gradient descent based methods – overfitting is often detected by observing the learning curves for training and validation error against the number of training steps. A typical behavior is that in the beginning of the training both training and validation loss are decreasing. This is the region where the model is still underfitting. At some point, when the model starts overfitting the training data, only the training loss decreases further while the validation loss starts to increase. The aforementioned early stopping method balances under- and overfitting by stopping the training once the validation loss starts to increase.

While in supervised learning one can easily select a well fit model by using the validation loss, in reinforcement learning one cannot apply this technique as the dataset is not fixed but dynamic and is constantly growing or changing. Furthermore, the quality of the current policy influences the quality of the data collected in the future. Even though learning a world model is in principle a supervised task, this problem also occurs in the model-based RL framework.

### 3.3 DYNAMIC UPDATE-TO-DATA RATIO

A typical hyperparameter in many RL algorithms is the update-to-data (UTD) ratio which specifies the number of update steps performed per environment step (i.e., per new data point). This ratio can in principle be used to balance under- and overfitting as one can control it in a way that not too few or too many updates steps are done on the currently available data. However, several problems arise while optimizing this parameter. First, it is very costly to tune this parameter as it requires to run the complete RL training several times making it infeasible for many potential applications. Second, the assumption that one fixed value is the optimal choice during the entire training duration does not necessarily hold. For example, if data from a newly explored region of the environment is added to the replay buffer it might be beneficial to increase the number of update steps.

To address these problems, we propose – DUTD – a new method that dynamically adjusts the UTD ratio during training. It is inspired by the early stopping criterion and targets at automatically balancing under- and overfitting online by adjusting the number of update steps. As part of the method, we store some of the experience in a separate validation buffer not used for training. Precisely, every $d$ environment steps we collect $s$ consecutive transitions from a few separate episodes dedicated to validation and every $k$ environment steps the world model is evaluated on the validation buffer, where $k$ should be much smaller than $d$. As the world model learning task is supervised this is easily done by recording the loss of the world model on the given validation sequences. The current validation loss is then compared to the validation loss of the previous evaluation. If the loss has decreased, we assume the model is still in the underfitting regime and increase the UTD rate by a specified amount. If the loss has increased, we assume the model to be in an overfitting regime and hence reduce the UTD rate. To allow for a finer resolution at the high-update side of the allowed interval we adjust the UTD rate in log-space, meaning it is increased or decreased by multiplying it with a value of $c$ or $1/c$ respectively, where $c$ is slightly larger than $1$. The update formula at time step $t$ then becomes

$$utd\_ratio_t = utd\_ratio_{t-k} \cdot b; \quad b = \begin{cases} c, & \text{if validation\_loss}_t < \text{validation\_loss}_{t-k}, \\ \frac{1}{c}, & \text{if validation\_loss}_t \geq \text{validation\_loss}_{t-k}. \end{cases} \quad (1)$$

DUTD is a general method that can be applied to any model-based RL algorithm that learns a world model in a supervised way. The implementation can be either in terms of the UTD ratio or the data-to-update ratio which is its inverse and which we call **IUTD** (i.e., the number of environment steps per update step). It is more convenient to use the UTD ratio if several updates are performed per environment step and the IUTD if an update step is only performed after some environment steps. Methodologically, the two settings are the same as the two ratios describe the same quantity and are just the inverse of each other.

A high-level overview of DUTD is shown in Figure 1 and the pseudocode is described in Algorithm 1, both explained in terms of the IUTD ratio as we will apply DUTD to the DreamerV2

algorithm Hafner et al. (2021) for which several update steps per environment step become computationally very costly. However, in both framework both scenarios can be addressed by letting the ratio be a fractional.

### 3.4 Applying DUTD to DreamerV2

We apply DUTD to DreamerV2 Hafner et al. (2021), which is a model-based RL algorithm that builds on Dreamer Hafner et al. (2020) which again builds on PlaNet Hafner et al. (2019). DreamerV2 learns a world model through latent imagination. The policy is learned purely in the latent space of this world model through an actor-critic framework. It is trained on imaginary rollouts generated by the world model. The critic is regressed onto $\lambda$-targets Schulman et al. (2015); Sutton & Barto (2018) and the actor is trained by a combination of Reinforce Williams (1992) and a dynamics backpropagation loss. The world model learns an image encoder that maps the input to a categorical latent state on which a Recurrent State-Space Model Hafner et al. (2019) learns the dynamics. Three predictors for image, reward, and discount factor are learned on the latent state. The total loss for the world model is a combination of losses for all three predictors and a Kullback–Leibler loss between the latents predicted by the dynamics and the latents from the encoder.

---

**Algorithm 1** DUTD (in terms of inverted UTD ratio)

---

**Input:** Initial *inverted* UTD ratio *iutd_ratio*; number of steps after which additional validation data is collected $d$, number of validation transitions collected $s$, steps after which the *iutd_ratio* is updated $k$, iutd update increment $c$

**for** $t = 1$ **to** total_num_of_env_steps **do**
    Act according to policy $\pi(a \mid s)$ and observe next state
    **if** $t \bmod d == 0$ **then**
        Collect $s$ transitions and store experience in a separate validation buffer; increment $t = t + s$
    **end if**
    **if** $t \bmod iutd\_ratio == 0$ **then**
        Perform one training step of the transition model
    **end if**
    **if** $t \bmod k == 0$ **then**
        Compute model loss $L$ on validation dataset
        **if** $L \geq L_{previous}$ **then**      # Overfitting
            $iutd\_ratio = iutd\_ratio \cdot c$
        **else**        # Underfitting
            $iutd\_ratio = iutd\_ratio/c$
        **end if**
        $L_{previous} = L$
    **end if**
**end for**

---

To apply DUTD we evaluate the image reconstruction loss on the validation set. Other choices are also possible but we speculate that the image prediction is the most difficult and important part of the world model. One could also use a combination of different losses but then one would potentially need a scaling factor for the different losses. As we want to keep our method simple and prevent the need of hyperparameter tuning for our method, we employ the single image loss. The source code of our implementation is publicly available [1].

## 4 Experiments

We evaluate DUTD applied to DreamerV2 on the Atari 100k benchmark Kaiser et al. (2019) and the DeepMind Control Suite Tassa et al. (2018). For each of the two benchmarks we use the respective hyperparameters provided by the authors in their original code base. Accordingly, the baseline IUTD ratio is set to a value of 5 for the control suite and 16 for Atari which we also use as initial value for our method. This means an update step is performed every 5 and 16 environment steps respectively. For both benchmarks we set the increment value of DUTD to $c = 1.3$ and the IUTD ratio is updated every 500 steps which corresponds to the length of one episode in the control suite (with a frameskip of 2). Every $100,000$ steps DUTD collects $3,000$ transitions of additional validation data. We cap the IUTD ratio in the interval $[1, 15]$ for the control suite and in $[1, 32]$ for Atari. This is in principle not necessary and we find that most of the time the boundaries, especially the upper one, is not reached. A boundary below 1 would be possible by using fractions and doing several updates per environment step, but this would be computationally very expensive for DreamerV2. All other hyperparameters are reported in the Appendix. They were not extensively tuned and we observed that the performance of our method is robust with respect to the specific choices. The environment steps in all reported plots also include the data collected for the validation set.

---

[1] https://github.com/Nicolinho/dutd

Figure 2: Aggregated metrics over 5 random seeds on the 26 games of Atari 100k with 95% confidence intervals according to the method presented in Agarwal et al. (2021). The intervals are estimated by the percentile bootstrap with statified sampling. Higher mean, median, interquantile mean (IQM) and lower optimality gap are better.

The Atari 100k benchmark Kaiser et al. (2019) includes 26 games from the Arcade Learning Environment Bellemare et al. (2013) and the agent is only allowed $100,000$ steps of environment interaction per game, which are $400,000$ frames with a frame-skip of $4$ and corresponds to roughly two hours of real-time gameplay. The final performance per run is obtained by averaging the scores of 100 rollouts with the final policy after training has ended. We compute the human normalized score of each run as $\frac{\text{agent score}-\text{random score}}{\text{human score}-\text{random score}}$. The DeepMind Control Suite provides several environments for continuous control. Agents receive pixel inputs and operate with a frame-skip of 2 as in the original DreamerV2. We trained for 2 million frames on most environments and to save computation cost for 1 million frames if standard DreamerV2 already achieves its asymptotic performance well before that mark. The policy is evaluated every $10,000$ frames for 10 episodes. For both benchmarks, each algorithm is trained with 5 different seeds on every environment.

Our experiments are designed to demonstrate the following:

- The UTD ratio can be automatically adjusted using our DUTD approach
- DUTD generally increases performance (up to 300% on Atari100k) by learning an improved world model compared to the default version of DreamerV2
- DUTD increases the robustness of the RL agent with regard to learning-related hyperparameters
- DUTD is competitive with the best UTD hyperparameter found by an extensive grid search

## 4.1 PERFORMANCE OF DUTD COMPARED TO STANDARD DREAMERV2

For Atari100k, Figure 2 shows results aggregated over the 26 games with the method of Agarwal et al. (2021), where the interquantile mean (IQM) ignores the bottom and top 25% of the runs across all games and computes the mean over the remaining. The optimality gap describes the amount by which a minimal value of human level performance is not reached. In Figure 11 we present the learning curves for each environment. The results show that DUTD achieves a drastically stronger performance on all considered metrics compared to DreamerV2 with the fixed default IUTD ratio of 16. It increases the interquantile mean (IQM) score by roughly $300\%$ and outperforms the human baseline in terms of mean score without any data augmentation.

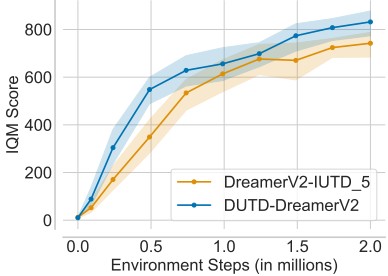

Figure 3 shows the aggregated results for two million frames over ten environments of the Control Suite, which we list in the Appendix. The curves per environment are presented in Figure 11 of the Appendix further including results for ten more environments on which the algorithms run until one million frames. Compared to the manually set default UTD ratio, DUTD matches or improves the performance on every environment. Overall, DUTD improves the performance significantly although its average IUTD rate over all games and checkpoints is $5.84$ similar to the default rate of $5$ showing that DUTD better exploits the performed updates.

Figure 3: Sample efficiency curves aggregated from the results for ten environments of the DeepMind Control Suite for DreamerV2 with the default UTD ratio and when it is adjusted with DUTD. The IQM score at different training steps is plotted against the number of environment steps. Shaded regions denote pointwise 95% stratified bootstrap confidence intervals according to the method by Agarwal et al. (2021).

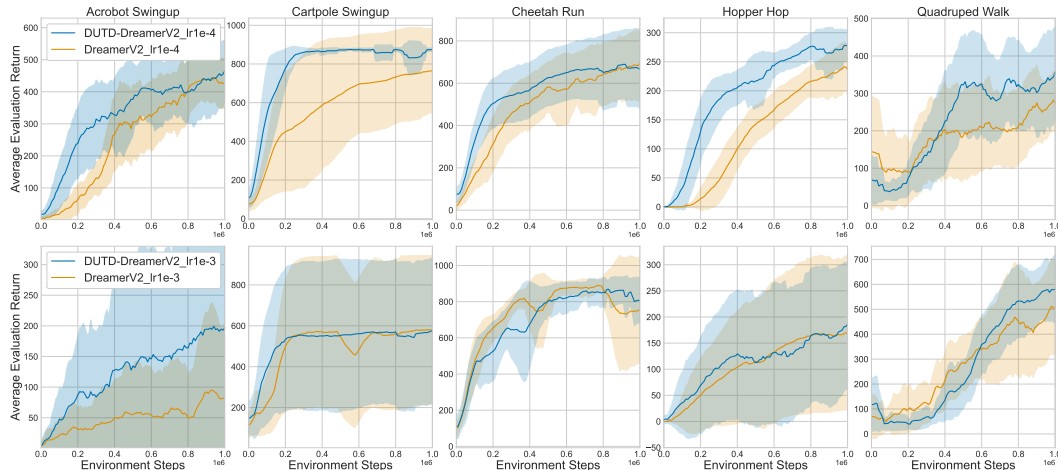

Figure 4: Learning curves for five environments of the Control Suite for DUTD-DreamerV2 and standard DreamerV2 when non-default learning rates are used. The first row shows the results for a lower than default learning rate of $0.0001$ and the second row for a higher one of $0.001$. The default learning rate is $0.0003$ and its results are shown in Figure 12. The solid line represents the mean and the shaded region a pointwise standard deviation in each direction computed over $5$ runs.

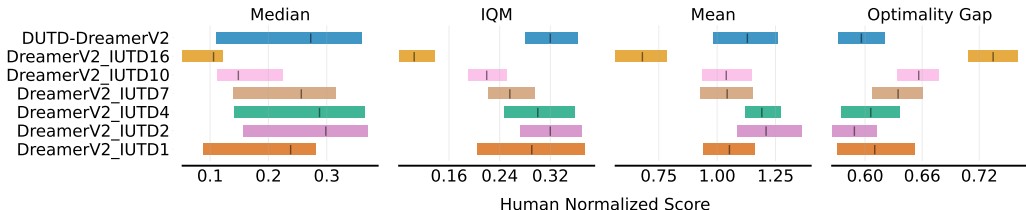

Figure 5: Aggregated metrics over $5$ random seeds on the $26$ games of Atari $100k$, cf. Figure 2 for the methodology. DUTD is compared to Dreamer with different choices for the IUTD rate.

## 4.2 INCREASED ROBUSTNESS WITH DUTD

As DUTD dynamically adjusts the UTD ratio which allows to modify the training process online, we formed the hypothesis that with DUTD the underlying RL algorithm is more robust to suboptimal learning hyperparameters. Similar to supervised learning on a fixed dataset the optimal number of updates to tradeoff between under- and overfitting will be highly dependent on hyperparameters like the learning rate. To investigate this, we evaluated DreamerV2 with and without our method for different learning rates of the dynamics model. The standard learning rate on the control suite is $0.0003$. Hence, we trained with both a higher learning rate of $0.001$ and a lower one of $0.0001$ on a subset of the environments. The resulting learning curves are displayed in Figure 4. Compared to the default learning rate the performance of DreamerV2 with the standard fixed IUTD ratio of $5$ is overall lower and decreases substantially for some of the environments for both non-default learning rates. However, using DUTD the algorithm achieves considerably stronger results. This shows that using DUTD the algorithm is more robust to the learning rate, which is an important property when the algorithm is applied in real world settings such as robotic manipulation tasks, since multiple hyperparameter sweeps are often infeasible in such scenarios. The need for more robustness as offered by DUTD is demonstrated by the performance drop of DreamerV2 with a learning rate differing by a factor of 3 and the fact that on Atari a different learning rate is used.

## 4.3 COMPARING DUTD WITH EXTENSIVE HYPERPARAMETER TUNING

In the previous sections, we showed that DUTD improves the performance of DreamerV2 with its default IUTD ratio significantly. Now we want to investigate how well DUTD compares to the best hyperparameter value for IUTD that can be found through an extensive grid search on each benchmark. While for many applications such a search is not feasible we are interested in what can be expected of DUTD relative to what can be regarded as the highest achievable performance.

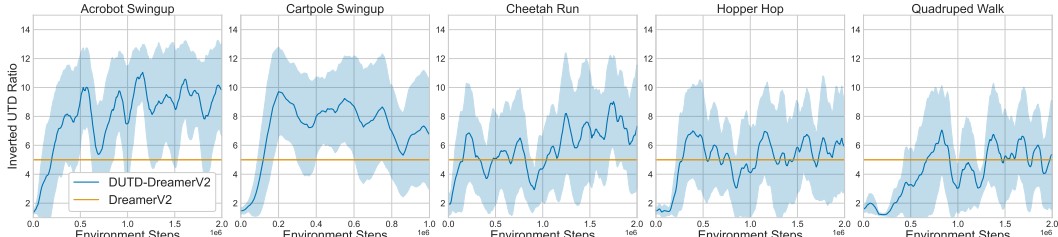

Figure 7: IUTD ratio against environment steps for DUTD and the standard DreamerV2 on five environments. For each environment the mean over 5 runs is plotted as the solid line and the shaded region represents one pointwise standard deviation in each direction.

On the Atari 100k benchmark we evaluate DreamerV2 with IUTD rates of $1, 2, 4, 7, 10$ and $16$ (the default value) and denote the algorithms with DreamerV2-IUTD_1, DreamerV2-IUTD_2, etc. The aggregated results over all games and seeds in Figure 5 show an increase in performance when the number of updates increases up to an IUTD rate of $2$. Increasing it further to $1$ leads to declining results. Thus, there is a sweet spot and one can not simply set the IUTD rate very low and expect good results. Averaged over all runs and checkpoints the IUTD rate of DUTD is at $3.91$ which is in the region of the best performing hyperparameters of $2$ and $4$. This is also reflected by the fact that DUTD achieves similar performance to these two optimal choices.

We further evaluate DreamerV2 with IUTD ratios of $2$, $5$ (the default one), $10$, and $15$ on ten environments of the control suite. An IUTD value below $2$ is not possible as a single run would take roughly two weeks to run on our hardware. The aggregated sample efficiency curves in Figure 6 further support the hypothesis that DUTD is competitive with the results of an extensive grid search. Only an IUTD choice of $2$ gives slightly better sample efficiency but reaches a lower final performance. To further investigate the behaviour of DUTD we report the adjusted **inverted** UTD ratio over time for five environments in Figure 7, and for all environments in Figure 13 in the Appendix. Interestingly, the behavior is similar for all the environments. At the start of the training, the ratio is very low and then it quickly oscillates around a value of roughly $5$ for most environments and an even higher value for a few others. On cheetah_run and hopper_hop, the IUTD oscillates around the default value of $5$ most of the time and still, DUTD reaches a higher performance than Dreamer as can be seen in the single environment plot in Figure 12 of the Appendix. This result supports

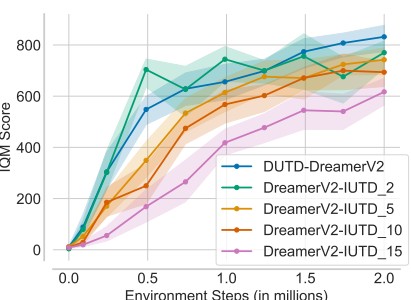

Figure 6: Sample efficiency curves showing the IQM score aggregated from the results for ten environments of the DeepMind Control Suite for DreamerV2 with different choices for the IUTD ratio. Shaded regions denote pointwise $95\%$ stratified bootstrap confidence intervals.

the hypothesis that a static IUTD rate can be suboptimal for some environments and that DUTD successfully balances over- and underfitting during the training process.

## 4.4 EVALUATION FOR A HIGH NUMBER OF SAMPLES

Next we investigate the behaviour of DUTD if training is continued for many environment steps. We randomly selected $5$ games from the Atari benchmark and trained the algorithms for $40$ million frames. The resulting learning curves displayed in Figure 8 show that DUTD maintains its advantage also in this setting. The significantly improved performance is achieved with an IUTD ratio of $13.51$ averaged over all games and checkpoints. In Figure 14 of the Appendix we show the development of the IUTD ratio over time for each environment. We can see that with DUTD after an initial phase with a lower IUTD ratio it oscillates around a value not too far from the highly tuned default ratio of $16$. This means DUTD significantly improves performance over plain DreamerV2 without requiring substantially more updates. The experiment further highlights the benefits of DUTD. Evaluating different choices for a fixed IUTD ratio in this setting is highly expensive and for low values of the IUTD ratio almost impossible as a single run with the default value takes already several days

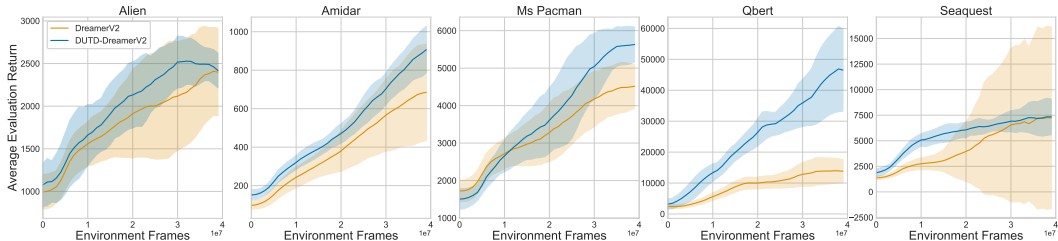

Figure 8: Learning curves for DreamerV2 with and without DUTD on 5 randomly selected environments of the Atari benchmark. For each environment the mean over 3 runs is plotted as the solid line and the shaded region represents one pointwise standard deviation in each direction.

to train. DUTD improves upon the highly tuned default choice and removes the need to tune this hyperparameter in an inner loop.

## 4.5 GENERALITY OF DUTD

To demonstrate the generality of DUTD we applied it to PlaNet Hafner et al. (2019) which is another model-based RL algorithm. We evaluated the resulting method on three environments of the DeepMind Control Suite using the same hyperparameters for DUTD as for Dreamer. The results in Figure 9 of the appendix show that DUTD also improves the performance of PlaNet validating that DUTD is a general method and indicating its usefulness for different base algorithms.

## 5 DISCUSSION

We presented a novel and general method denoted as DUTD that is designed to detect under- and overfitting on evolving datasets and is able to dynamically adjust the typically hand-set UTD ratio in an automated fashion. As in early stopping, the underlying rationale is that too many updates can lead to overfitting while too few updates can lead to underfitting. DUTD quickly identifies such trends by tracking the development of the world model performance on a validation set. It then accordingly increases or decreases the UTD ratio in the case of underfitting or overfitting.

In our experiments, we demonstrated how to successfully apply DUTD to a model-based RL algorithm like DreamerV2. The experiments show that DUTD can automatically balance between the under- and overfitting of the world model by adjusting the UTD ratio. As a result, DUTD removes the burden of manually setting the UTD ratio, which otherwise needs to be tuned for new environments making it prohibitively expensive to apply such algorithms in many domains. At the same time, DUTD increases the performance of DreamerV2 significantly compared to its default UTD rate and is competitive with the best hyperparameter found for each domain through an extensive hyperparameter search. Moreover, a notable property of DUTD-DreamerV2 is its robustness to changes in the learning rate. This is important, as the learning rate often has to be tuned for new environments. For example, in DreamerV2 the default learning rate differs between Atari and the DeepMind Control Suite. In the context of real world problems such tuning is undesirable and often too costly. At the same time, the hyperparameters of DUTD can easily be set and do not have a big influence on the final performance. We recommend updating the UTD rate after a fixed time interval that is similar to the average episode length. The data used for validation should not exceed 10% of all data.

An interesting avenue for future work would be to explore non-supervised objectives for model-free RL algorithms that can be used for evaluation on the validation set. This would allow the usage of DUTD to adjust the UTD ratio of such algorithms. Another potential way to further boost the performance of our method is to use k-fold cross-validation with an ensemble of world models such that every transition can be used for training.

We are convinced that DUTD is a further step in the direction of autonomy and the easy applicability of RL algorithms to new real world problems without the need to tune any hyperparameters in an inner loop. More generally, our work shows that it might be fruitful to use knowledge about the underlying learning dynamics to design algorithms that dynamically adjust parts of the learning algorithm.

ACKNOWLEDGMENTS

This work was supported by the European Union's Horizon 2020 Research and Innovation Program under Grant 871449-OpenDR.

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

## A FURTHER RESULTS

### A.1 APPLYING DUTD TO PLANET

To demonstrate the generality of DUTD we additionally applied it to PlaNet Hafner et al. (2019) with the same hyperparameters for DUTD as we also used for DreamerV2. As base source code on which we implemented DUTD we used Pineda et al. (2021). We evaluated the resulting algorithm on three environments of the DeepMind Control Suite that were also used in the original publication of PlaNet. We used 5 seeds and evaluated the algorithms every 25000 environment frames. The results in Figure 9 show that DUTD also improves the performance of PlaNet. This is further evidence for the generality of DUTD.

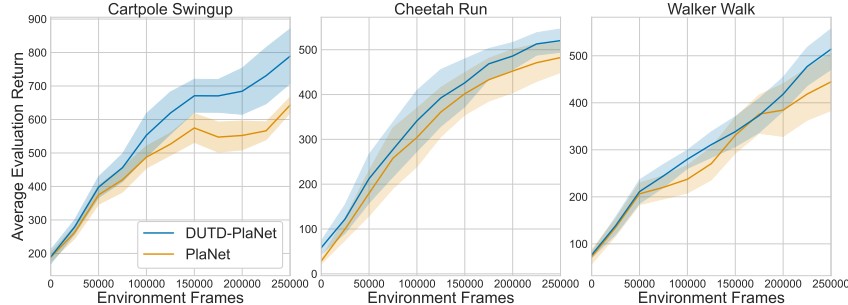

Figure 9: Learning curves for PlaNet with and without DUTD on three environments of the Deep-Mind Cotrol Suite. The solid line is the mean over 5 seeds and the shaded area represents one pointwise standard deviation. We used a uniform filter of size 3.

### A.2 DETAILED RESULTS FOR APPLYING DUTD TO DREAMERV2

The ten environments of the DeepMind Control Suite used to generate the aggregated curves in the Figures 3 and 6 are: acrobot_swingup, cheetah_run, finger_turn_easy, finger_turn_hard, hopper_hop, quadruped_run, quadruped_walk, reacher_hard, walker_walk, and walker_run.

We evaluated on all 20 environments used in the original Dreamer paper Hafner et al. (2020) but to save computation stopped training for ten environments at 1 million steps because standard Dreamer already reaches its asymptotic performance well before that mark. The aggregated curves are generated from the other 10 environments for which training ran until 2 million steps. Figure 12 shows the single learning curves for all environments. Please note, that on the 1 million steps environments with DUTD the asymptotic performance is reached much faster - often twice as fast.

In the Figures 10, 11, 12, 13, and 15 we present the more detailed results of our experiments for each single environment.

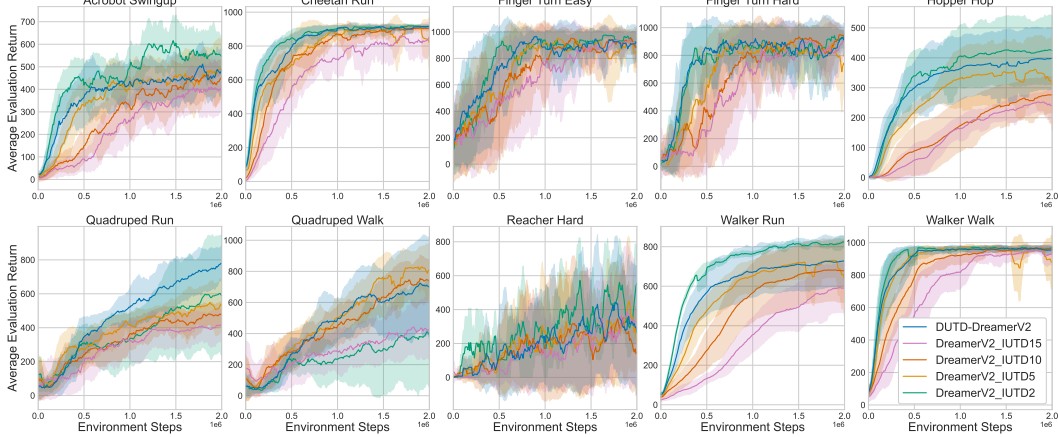

Figure 10: Learning curves for different choices of the IUTD ratio for each of the environments. The solid line is the mean over 5 seeds and the shaded area represents one pointwise standard deviation.

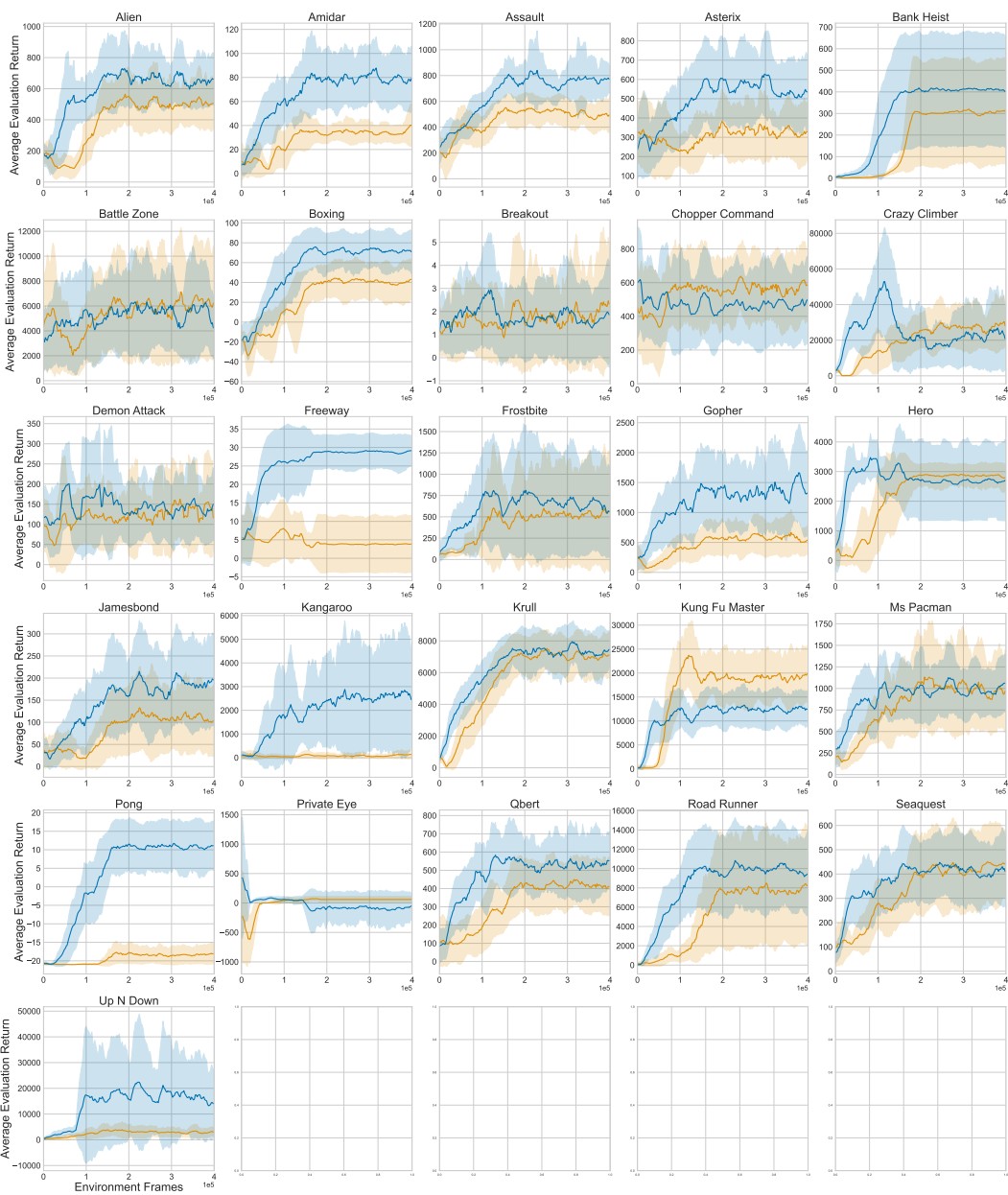

Figure 11: Learning curves for DreamerV2 with and without DUTD on the 26 environments of the Atari 100k benchmark. The solid line is the mean over 5 seeds and the shaded area represents one pointwise standard deviation.

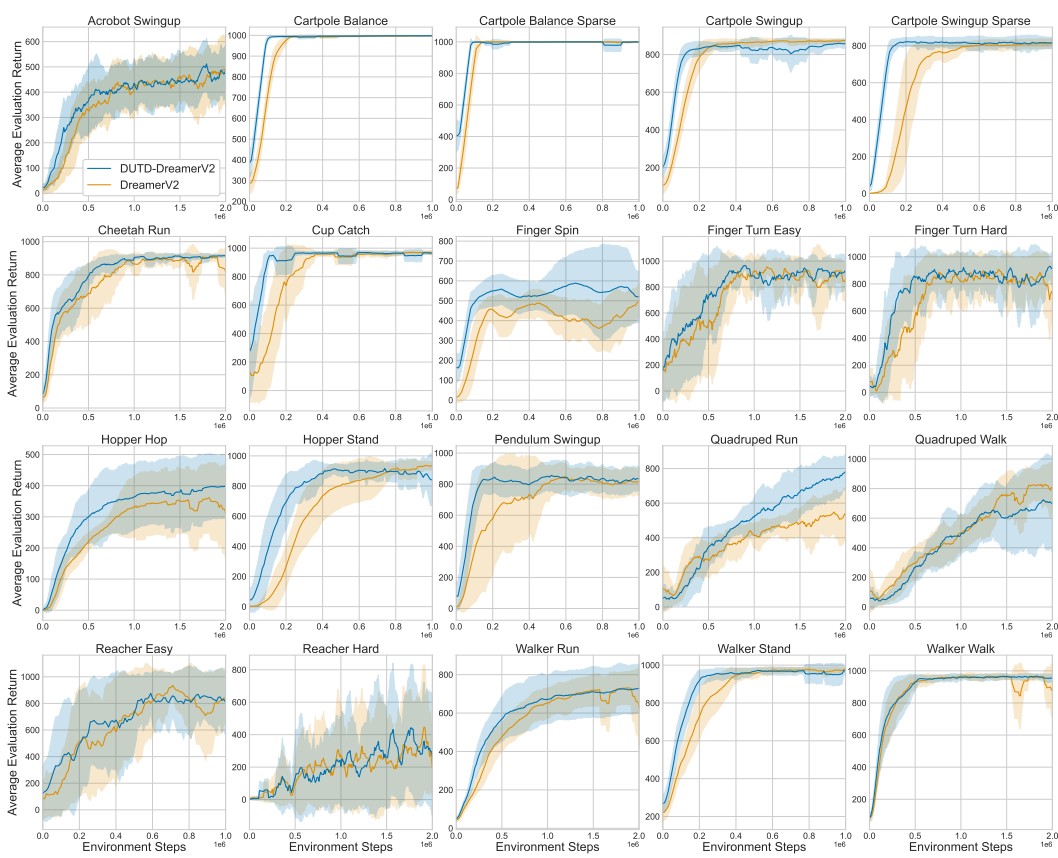

Figure 12: Learning curves for DreamerV2 with and without DUTD for 20 environments of the DeepMind Control Suite. The solid line is the mean over 5 seeds and the shaded area represents one pointwise standard deviation.

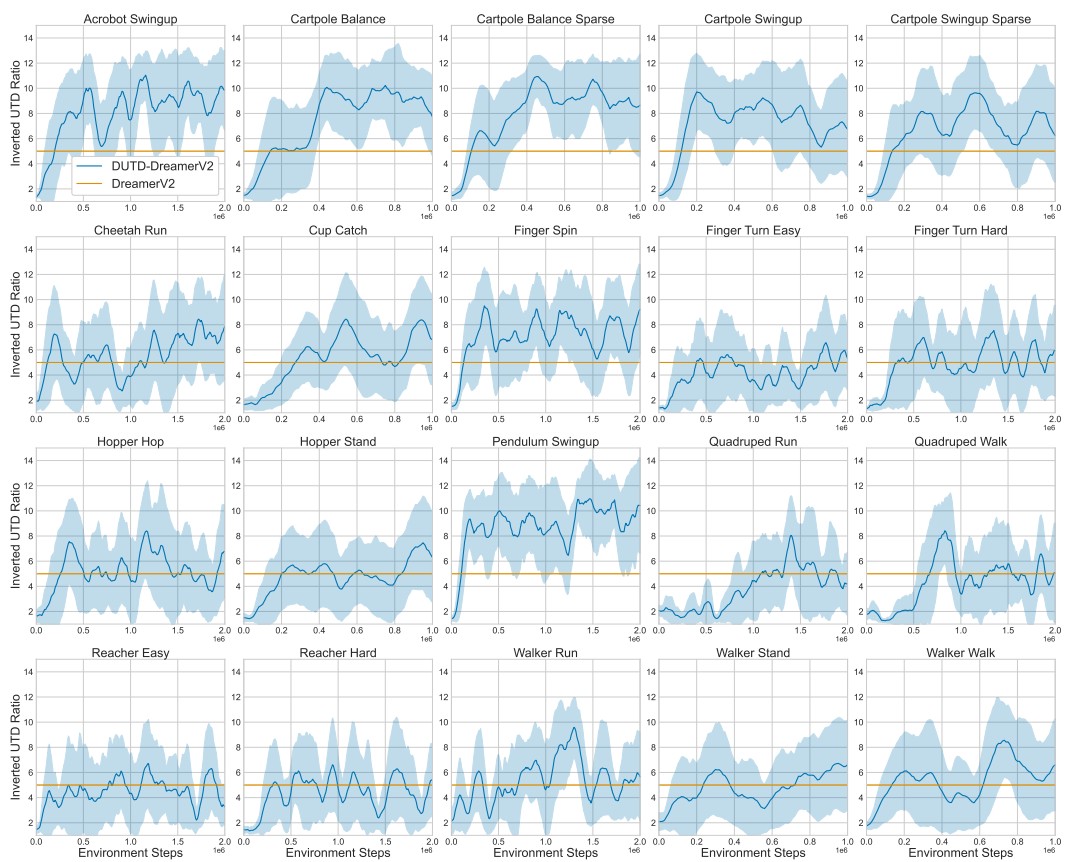

Figure 13: IUTD ratio against environment steps for DUTD and the standard DreamerV2 on all environments. For each environment the mean over 5 runs is plotted as the solid line and the shaded region shows represents one pointwise standard deviation in each direction.

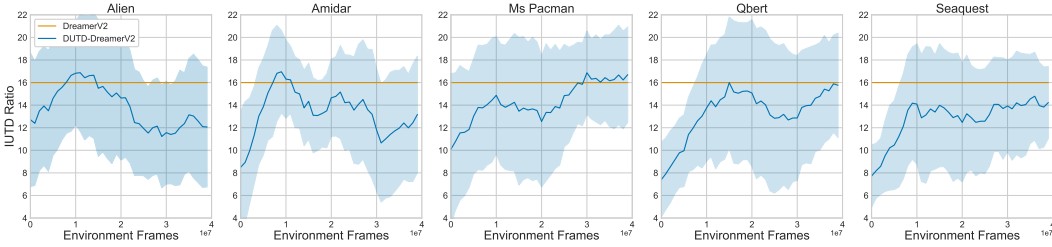

Figure 14: IUTD ratio against environment steps for DUTD and the standard DreamerV2 on 5 environments of Atari for which the algorithms were trained until 40 million frames. For each environment the mean over 3 runs is plotted as the solid line and the shaded region shows represents one pointwise standard deviation in each direction.

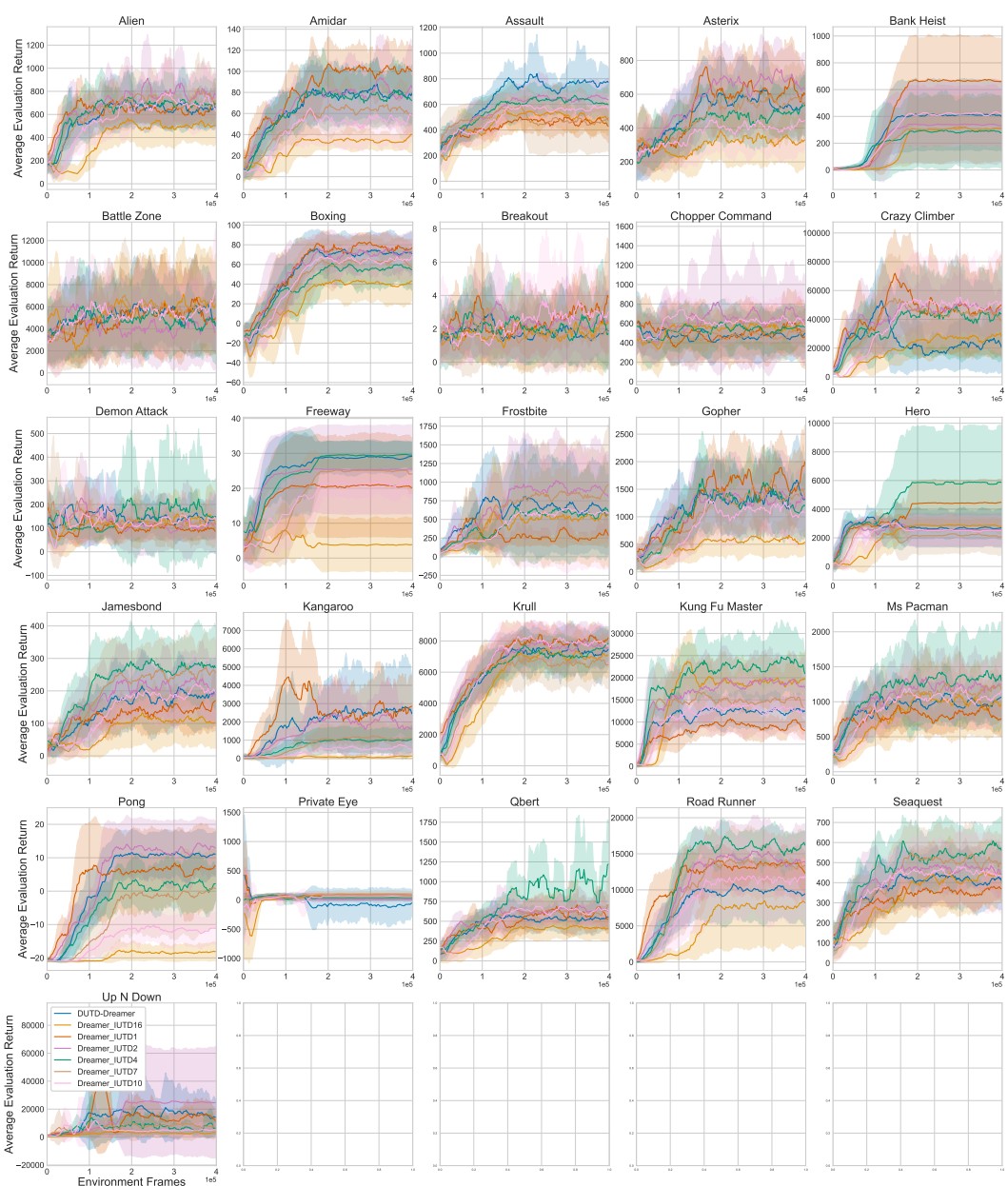

Figure 15: Learning curves for different choices of the IUTD ratio for each of the 26 environments of the Atari 100k benchmark. The solid line is the mean over 5 seeds and the shaded area represents one pointwise standard deviation.

# B  HYPERPARAMETERS

In Table 1 we give an overview of all hyperparameters related to DUTD. All other hyperparameters are the standard DreamerV2 hyperparameters as given in the open source codebase [2]. On the DM Control Suite we reduced the number of steps $d$ after which to collect new data for the validation set by a half during the first 400k steps as for some environments a strong policy is learned very quickly and hence a validation set with more recent transitions that better represent the kind of transitions the agent encounter makes more sense. We have because we started our first experiments with this but from some limited additional experiments it seems not to have a big impact on performance.

Table 1: Hyperparameters values for DUTD applied to DreamerV2 and the corresponding hyperparameter in the original DreamerV2.

| HYPERPARAMETER | ATARI | DM CONTROL |
|---|---|---|
| INITIAL IUTD RATIO | 16 | 5 |
| LOWER BOUNDARY FOR THE IUTD RATIO | 1 | 1 |
| UPPER BOUNDARY FOR THE IUTD RATIO | 32 | 15 |
| IUTD UPDATE INCREMENT — $c$ | 1.3 | 1.3 |
| NUMBER OF STEPS AFTER WHICH TO UPDATE THE IUTD RATIO — $k$ | 500 | 500 |
| VALIDATION SET MAXIMUM SIZE — $k$ | 12,000 | 10,000 |
| NUMBER OF STEPS AFTER WHICH TO COLLECT NEW DATA FOR THE VALIDATION SET — $d$ | 100,000 | 100,000 |
| NUMBER OF ADDITIONAL TRANSITIONS FOR THE VALIDATION SET EACH TIME NEW VALIDATION DATA IS COLLECTED — $s$ | 3,000 | 3,000 |
| | STANDARD DREAMERV2 | |
| IUTD RATIO | 16 | 5 |

# C  HYPERPARAMETER SENSITIVITY OF DUTD

Most hyperparameters of our method are straightforward to set and do not need any tuning. Updating the UTD ratio after the maximum episode length of 500 in DM Control Suite (DMC) is a value that we directly transferred to the Atari benchmark without further tuning. The initial value for the UTD ratio has no effect, as it gets quickly adjusted. The lower and upper limits for the UTD ratio are not reached often and hence do not affect performance given they are chosen lavish enough. We did not tune those. We tried a few choices for the number of additional transitions each time new validation data is collected and the number of steps after which we do so but did not find it to affect performance a lot and fixed one choice for both benchmarks.

The multiplicative factor $c$ is the most important hyperparameter of our method and we hence conducted an additional experiment evaluating its sensitivity on the Atari100k benchmark over 5 random seeds. We show the aggregated metrics for different multiplicative factors in Figure 16.

The results show that seen over all metrics and relative to the baseline results the performance is not very sensitive with respect to the choice of the multiplicative factor. For the mean our default factor of 1.3 even gives slightly worse results than all other factors. Further, we argue the fact that the same setting of hyperparameters of DUTD works for very different benchmarks, Atari and DMC, shows that DUTD is not very sensitive to its hyperparameters and that the default values given by us will most likely work for a wide range of tasks. While an extensive hyperparameter search for the optimal UTD ratio might give slightly better results than DUTD with some fixed multiplicative factor, DUTD is still favorable for many real world applications where such tuning is too costly.

---

[2] https://github.com/danijar/dreamerv2

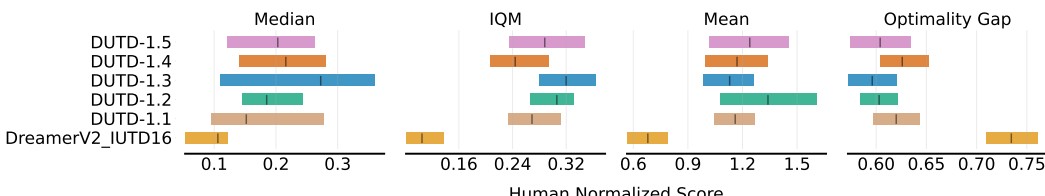

Figure 16: Aggregated metrics over 5 random seeds on the 26 games of Atari 100k, cf. Figure 2 for the methodology. We investigate the sensitivity of DUTD to its own most important hyperparameter $c$ for values of 1.1, 1.2, 1.3 (default one used in the main experiments), 1.4, and 1.5 .

