# OpenReview forum: "Dynamic Update-to-Data Ratio: Minimizing World Model Overfitting"
_ICLR.cc/2023/Conference — ICLR 2023 poster_

### Official Review · Reviewer_V4T4 · 2022-10-23

**Confidence:** 3
**Correctness:** 4
**Technical Novelty And Significance:** 2
**Empirical Novelty And Significance:** 3
**Recommendation:** 6

**Clarity, Quality, Novelty And Reproducibility:**

In general, the paper is well presented and well written. But the novelty is somewhat limited.

**Strength And Weaknesses:**

Strength:
* The paper is well written and easy to follow.
* The proposed method is easy to use in practice.
* The experiments showed promising results with the DreamerV2 algorithm.

Weakness:

the novelty of the paper is limited. The proposed method is a heuristic method which does not add much theoretical contribution. The form of the UITD ratio update is intuitive but it is not clear why this particular multiplicative form is chosen among other possibilities.

The choice of c is not well explained or examined, I am also curious to know whether having an adaptive form of c that changes over time helps.

All the results are based on one model-based RL algorithm (DreamerV2).


**Summary Of The Paper:**

The paper takles the problem of overfitting in the RL setting, where early stopping is not directly applicable due to the evolution of the dataset. The proposed method adjusts the update to data (UTD) ratio during training which trades off between underfitting and overfitting. The experiments are conducted using a model-based RL algorithm (DreamerV2) on the DeepMind Control Suite and Atari 100k benchmark, which showed competitiave resutls compared with extensive hyperparameter search.



**Summary Of The Review:**

The paper is easy to follow and the proposed method leads to easy applicability of RL algorithms in real life, which will bring benefits especially to practitioners. However, the novelty of the paper is limited and it does not provide much insights into a principled way of trading off under- versus overfitting in RL problems.

---

> ### Author Response · Authors · 2022-11-17
> **Answer to Reviewer V4T4**
>
> We would like to thank the reviewer for their constructive reply and comments.
>
> > All the results are based on one model-based RL algorithm (DreamerV2).
>
> We agree with the reviewer that applying DUTD to additional model-based RL algorithms allows for a better assessment of the generality of DUTD. Hence, we conducted an additional experiment where we applied DUTD to the model-based RL algorithm PlaNet. We added a corresponding section to the experimental results in the manuscript. The results show that DUTD also improves the performance of PlaNet validating the generality of the proposed method.
>
> > The proposed method is a heuristic method which does not add much theoretical contribution.
>
> We would like to emphasize that both theoretical insights and algorithms that work in practice can be useful contributions to the field. Moreover, our method is well motivated by empirical and theoretical knowledge about under- and overfitting and its connection to the number of gradient updates.
> Our work further highlights the importance of taking the dynamic nature of the learning problem in (model-based) RL into account. We are convinced this insight opens the door for many improved algorithms by solving challenges originating from this dynamic nature in a similar manner as we did in our work.
>
>
> > The form of the UITD ratio update is intuitive but it is not clear why this particular multiplicative form is chosen among other possibilities.
>
> We also experimented with an additive update rule but found the multiplicative one, which corresponds to tuning in log instead of linear space, to be more beneficial. For example, with an additive constant of 2 the  IUTD ratios of 1 and 3 would be just one step away while they have very different effects on the training while IUTD ratios of 20 and 22 would also be just one step away while they produce almost the same training behavior. Adjusting the ratio in log space (i.e. through multiplication) allows for a finer resolution at the high-update side of the scale and a coarser resolution at the low-update side.
>
> > The choice of c is not well explained or examined
>
> We would like to highlight that we provided a sensitivity analysis for the value of $c$ in Figure 16 (Figure 15 before the revision) of the appendix.
>
> > I am also curious to know whether having an adaptive form of c that changes over time helps.
>
> We agree with the reviewer that in principle it might be possible to improve the algorithm by having a schedule for $c$. However, it is not clear to us how such a schedule might look like. Further, this would make the method more complex and potentially introduce additional hyperparameters. At the same time, we happily embrace possible future extensions of our work.

---

### Official Review · Reviewer_uze4 · 2022-10-24

**Confidence:** 4
**Correctness:** 4
**Technical Novelty And Significance:** 3
**Empirical Novelty And Significance:** 4
**Recommendation:** 8

**Clarity, Quality, Novelty And Reproducibility:**

* clearly written paper
* no concerns about reproducibility
* thorough empirical evaluation
* to the best of my knowledge this is the first paper introducing a variable UTD ratio.

**Strength And Weaknesses:**

Strengths:
* simple idea that is relatively easy to implement and could potentially be extended to other settings as well.
* clear, well-written paper with extensive empirical evaluation.
* promising empirical results. The fact that the approach is competitive with a UTD ratio tuned by grid search is particularly promising because tuning the UTD is often very costly and time-consuming in practice.

Weaknesses:
* in appendix B there seems to be some special casing for the control suite. This detracts somewhat from the simplicity of the methods. Can you show empirically what happens when this is removed?

**Summary Of The Paper:**

This paper looks at overfitting in world model learning. It proposes a mechanism to dynamically update the update to data ratio during training. This can be seen as an analogue of early stopping in supervised learning. The paper introduces a small validation replay buffer used for adjusting the UTD ratio and adapts it based on the target validation loss. The paper evaluates the approach building on the dreamerv2 method. Empirically this approach is competitive with dreamerv2 with a tuned UTD ratio.

**Summary Of The Review:**

Nice paper that thoroughly evaluates a simple idea to dynamically update the UTD in world model based algorithms and shows strong empirical results.

---

> ### Author Response · Authors · 2022-11-17
> **Answer to Reviewer uze4**
>
> We would like to thank the reviewer for their thoughtful and insightful review.
>
> > in appendix B there seems to be some special casing for the control suite. This detracts somewhat from the simplicity of the methods. Can you show empirically what happens when this is removed?
>
> We would like to point out that the few different hyperparameters are either an artifact of the benchmark or do not have a strong impact on the performance:
> - initial IUTD ratio: we used the default fixed IUTD ratio of Dreamer for the respective benchmark. As it changes immediately anyway it also has no big effect on the behavior of the algorithm.
> - Upper boundary for the IUTD ratio: the lower maximal IUTD in DMC has no effect as it is never reached, as can be seen in Figure 12 (13 in the revision) in the Appendix. We set it in the beginning of the project to a multiple of the default IUTD ratio in Dreamer and never changed it, but the results would have been the same if the value of 32 would have been used also in DMC.
> - Validation set maximum size:  For Atari100K we used 12000 as this is the total amount of validation transition collected in this benchmark. For DMC we used the more 'reasonable' value of 10000. They are very similar and it should not make a noticeable difference in performance. The different values are not the result of prior experimentation.

---

### Official Review · Reviewer_U8TF · 2022-10-25

**Confidence:** 3
**Correctness:** 4
**Technical Novelty And Significance:** 3
**Empirical Novelty And Significance:** 4
**Recommendation:** 8

**Clarity, Quality, Novelty And Reproducibility:**

- The method, primary contributions, and demonstration of effectiveness is presented clearly.
- The experiments demonstrate the high quality of their method for automatically tuning the UTD parameter and the importance of this choice.
- The idea of using a validation set to identify overfitting and perform early stopping is not novel, but their method for applying this concept to MBRL is new as far as I know.
- Their thorough description of their method, hyperparameters and results, and their use of multiple seed so provide error bars makes this work easily reproducible, as will their code release.

Small Note:
- In the final paragraph in Section 2, missing word: “In contrast to these approaches we [present] our method”

**Strength And Weaknesses:**

Strengths:
- The method, though intuitive, is not standard practice and they make strong case for why DUTD could benefit all MBRL practitioners by improving their training performance while reducing the amount of manual tuning necessary.  Such improvements also enable fairer comparisons between future methods by removing one of many possible confounding factors in their evaluation.
- Thorough experimentation and presentation of findings, including their many plots in the Appendix supports both drawing additional conclusions and reproducibility. Reporting results using methods described in Agarwal et al 2021 supports fair comparisons and a stronger ability to draw meaningful conclusions.
- Not mentioned in the work, but the method also allows for the UTD parameter to be adaptively set for each individual task automatically, with different tasks using substantially different average values (as seen in Figure 12).

Weaknesses:
- These results are only evaluated on DreamerV2, therefore it is unclear if they extend to other MBRL methods or are Dreamer specific.
- The claim that DUTD is more robust to hyper parameters than a fixed value of UTD is supported by comparing DUTD to the default fixed value of UTD which was already demonstrated to be sub-optimal.  Since it was already demonstrated that DUTD outperforms the default UTD with the default hpms then it is unsurprising that this continues to be the case when changing the hpms.  To isolate whether the adaptive choice of UTD as done by DUTD is more robust to varying hpms (the goal of this ablation), it should be compared to the best fixed UTD value as chosen by careful tuning.
- Similarly, it seems a possible takeaway from the longer runs is the value of adaptive UTD setting as we see in Figure 13 that the value chosen by DUTD varies throughout training.  However, because the comparison in Figure 8 is to the default value instead of the best tuned value of UTD, it is unclear whether there is benefit from adaptively setting UTD throughout training or just from choosing a good value of UTD.

**Summary Of The Paper:**

This paper aims to improve the training of Model Based Reinforcement Learning methods by introducing an automatic method of setting/dynamically tuning the update-to-data (UTD) ratio, a key hyper parameter which controls how much the world model over- or under-fits the training data.  They propose using a held out set of validation samples, updated throughout training, to estimate whether the world model is over- or under-fitting and to update the UTD ratio accordingly.  This method both enables the parameter to be set automatically, avoiding the need for hyper parameter tuning, and also allows the parameter to be dynamically updated as the training stage evolves.  They present results using their approach with DreamerV2 on both the DeepMind Control Suite and the Atari100k benchmark, demonstrating comparable performance on Atari100k and a slight improvement on DeepMindControl Suite when compared to training with the best fixed value of UTD when it is carefully tuned.  Additionally they show that the final performance on both environment is highly dependent on carefully tuning UTD when using a fixed value, but the performance of their method is robust to a wide range of values for its primary hyper-parameter.

**Summary Of The Review:**

This paper (1) provides strong evidence that tuning the UTD parameter substantially impacts the outcomes of DreamerV2 training (2) provides a method for automatically tuning the UTD parameter that consistently matches or improves the performance of the best fixed value of UTD when chosen with careful tuning.

This method is likely broadly applicable to MBRL practitioners and can be used as a standard tool for effectively training world models, saving time/resources in hyper parameter tuning and making results more directly comparable by removing the confounding factor of how well tuned the UTD parameter is.  However, it is not clear that this is the case since the method has only been evaluated on one such method, DreamerV2 (my main concern when making my acceptance recommendation).  Additionally, there isn’t clear evidence for the claim that *adaptively* setting UTD throughout training is more beneficial than choosing the optimal fixed UTD (my main concern when evaluating correctness at 3 out of 4).

---

> ### Author Response · Authors · 2022-11-17
> **Answer to Reviewer U8TF**
>
> We would like to thank the reviewer for their review and suggestions.
>
> > These results are only evaluated on DreamerV2, therefore it is unclear if they extend to other MBRL methods or are Dreamer specific.
>
> We agree with the reviewer that applying DUTD to additional model-based RL algorithms allows for a better assessment of the generality of DUTD. Hence, we conducted an additional experiment where we applied DUTD to the model-based RL algorithm PlaNet. We added a corresponding section to the experimental results in the manuscript. The results show that DUTD also improves the performance of PlaNet validating the generality of the proposed method.
>
> > The claim that DUTD is more robust to hyper parameters than a fixed value of UTD is supported by comparing DUTD to the default fixed value of UTD which was already demonstrated to be sub-optimal. Since it was already demonstrated that DUTD outperforms the default UTD with the default hpms then it is unsurprising that this continues to be the case when changing the hpms. To isolate whether the adaptive choice of UTD as done by DUTD is more robust to varying hpms (the goal of this ablation), it should be compared to the best fixed UTD value as chosen by careful tuning.
>
> Unfortunately, it is computationally too demanding to have baselines with different UTDs for every experiment. More importantly, however, we do not claim and it is also not the purpose of the mentioned experiment that DUTD achieves a stronger performance than an extensive hyperparameter search. Rather, we want to demonstrate the usefulness of DUTD for many real-world settings where a single training run is very costly and neither the optimal learning rate nor the optimal UTD ratio are known beforehand.  The experiment shows that with our adaptive UTD ratio the algorithm is robust to different choices of the learning rate. Hence, with DUTD two important hyperparameters do not need to be tuned in an inner loop while still obtaining a reasonably strong performance.
>
> > it seems a possible takeaway from the longer runs is the value of adaptive UTD setting as we see in Figure 13 that the value chosen by DUTD varies throughout training. However, because the comparison in Figure 8 is to the default value instead of the best tuned value of UTD, it is unclear whether there is benefit from adaptively setting UTD throughout training or just from choosing a good value of UTD
>
> Using very low IUTD ratios is not feasible in this setting as a single training run would already take several weeks. Moreover, it is very likely that the default value in Dreamer is already well-tuned for this setting as it is the setting Dreamer was evaluated on in the original publication. As shown in Figure 5, a too small IUTD ratio leads to decreasing performance. In the high-sample and hence many total updates regime it is very likely that the threshold where more updates hurt performance is much higher than in the Atari100k setting.
>
> > Additionally, there isn’t clear evidence for the claim that adaptively setting UTD throughout training is more beneficial than choosing the optimal fixed UTD (my main concern when evaluating correctness at 3 out of 4)
>
> We would like to clarify that we do not claim that adapting the UTD ratio during training is more beneficial than choosing the optimal fixed UTD ratio. Instead, we claim that the adaptive UTD ratio is better than the default hyperparameter value and competitive with an extensive hyperparameter search, which roughly corresponds to the optimal fixed UTD ratio.

---

> > ### Comment · Reviewer_U8TF · 2022-11-21
> > **Reviewer Response**
> >
> > Thank you for the answers and additional experiments, I find the PlaNet results resolve my concerns about the method only being tested on a single algorithm and I appreciate the clarification of your claims.  In light of this I have raised my score, I find this to be a meaningful contribution to the field.

---

> > > ### Author Response · Authors · 2022-11-22
> > > **Author Response**
> > >
> > > Thank you for helping to increase the quality of the paper and for raising your score.

---

### Decision · Program_Chairs · 2023-01-20

**Decision:**

Accept: poster

**Justification For Why Not Higher Score:**

The idea seems like a small adjustment of the existing algorithms.

**Justification For Why Not Lower Score:**

The idea is simple but seems to work well.

**Metareview: Summary, Strengths And Weaknesses:**

In the context of model-based reinforcement learning (MBRL), the paper proposes an automatic method of adjusting the update-to-data (UTD) ratio, a hyperparameter which controls how many updates is done to the world model between collecting new data. The method is based on using a validation set to detect model over-/underfitting and adjusting the UTD ratio accordingly. Experiments with the DreamerV2 and PlaNet models on the DeepMind Control Suite and the Atari100k benchmark suggest that the proposed adjustment yields performance competitive with the performance of the same model with a careful tuned value of the UTD ratio hyperparameter.

During the discussion, the reviewers raised the following concerns:
- The proposed trick was originally evaluated only on DreamerV2. The authors provided extra experiments showing that the technique also works on PlaNet.
- The novelty of the paper is limited: the proposed technique is a relatively small adjustment to the existing algorithms.

Despite the concerns, the proposed idea is simple but effective and it can become a useful tool for MBRL practitioners.

**Note From Pc:**

if the above contains the word "oral" or "spotlight" please see: "oral" presentation means -> notable-top-5% and "spotlight" means -> notable-top-25%. As stated in our emails, we are disassociating presentation type from AC recommendations